# Prospective Evaluation of Resection Margins Using Standardized Specimen Protocol Analysis among Patients with Distal Cholangiocarcinoma and Pancreatic Ductal Adenocarcinoma

**DOI:** 10.3390/jcm10153247

**Published:** 2021-07-23

**Authors:** Jonathan Garnier, Jacques Ewald, Flora Poizat, Eddy Traversari, Ugo Marchese, Anais Palen, Jean Robert Delpero, Olivier Turrini

**Affiliations:** 1Department of Surgical Oncology, Institut Paoli-Calmettes, 13009 Marseille, France; ewaldj@ipc.unicancer.fr (J.E.); traversari.eddy@gmail.com (E.T.); u.marchese@hotmail.fr (U.M.); palena@ipc.unicancer.fr (A.P.); 2Department of Pathology, Institut Paoli-Calmettes, 13009 Marseille, France; poizatf@ipc.unicancer.fr; 3Department of Surgical Oncology, Institut Paoli-Calmettes, CRCM, Aix-Marseille University, 13009 Marseille, France; jrdelpero@numericable.fr (J.R.D.); turrinio@ipc.unicancer.fr (O.T.)

**Keywords:** distal cholangiocarcinoma, pancreatic ductal adenocarcinoma, margin, R1 resection

## Abstract

Purpose: Using a standardized specimen protocol analysis, this study aimed to evaluate the resection margin status of patients who underwent resection for either distal cholangiocarcinoma (DC) or pancreatic ductal adenocarcinoma (PDAC). This allowed a precise millimetric analysis of each inked margin. Methods: From 2010 to 2018, 355 consecutively inked specimens from patients with PDAC (*n* = 288) or DC (*n* = 67) were prospectively assessed. We assessed relationships between the tumor and the following margins: transection of the pancreatic neck, bile duct, posterior surface, margin toward superior mesenteric artery, and the surface of superior mesenteric vein/portal vein groove. Resection margins were evaluated using a predefined cut-off value of 1 mm; however, clearances of 0 and 1.5 mm were also evaluated. Results: Patients with DC were mostly men (64% vs. 49%, *p* = 0.028), of older age (68 yo vs. 65, *p* = 0.033), required biliary stenting more frequently (93% vs. 77%, *p* < 0.01), and received less neoadjuvant treatment (*p* < 0.001) than patients with PDAC. The venous resection rate was higher among patients with PDAC (*p* = 0.028). Postoperative and 90-day mortality rates were comparable. Patients with PDAC had greater tumor size (28.6 vs. 24 mm, *p* = 0.01) than those with DC. The R1 resection rate was comparable between the two groups, regardless of the clearance margin. Among the three types of resection margins, a venous groove was the most frequent in both entities. In multivariate analysis, the R1 resection margin did not influence patient survival in either PDAC or DC. Conclusion: Our standardized specimen protocol analysis showed that the R1 resection rate was comparable in PDAC and DC.

## 1. Introduction

Patients diagnosed with either non-metastatic distal cholangiocarcinoma (DC) or pancreatic ductal adenocarcinoma (PDAC) are managed with a common therapeutic strategy. For patients with resectable disease, upfront pancreaticoduodenectomy is performed as neoadjuvant treatment has not yet been validated. In patients with locally advanced disease, induction treatment is recommended first, eventually followed by resection in those who are fit and otherwise eligible. Despite common therapeutic strategies, a large series study published in 2017 [1] showed that DC and PDAC should not be considered the same entity, and that survival was better for patients who had DC, while PDAC patients had a higher rate of margin involvement (R1). Moreover, margin invasion was found to be an independent predictor of survival in both cancers. However, the consistency of margin status assessment in that study is questionable. Since 2009, the Royal College of Pathologists and the Leeds Pathology Protocol [2] have provided guidelines, updated in 2019 [3], to assess margin status among patients who undergo pancreaticoduodenectomy for pancreatic head malignancies. Several studies have shown the relevance of this specimen analysis technique [4,5], and it is currently recommended by both the Royal College of Pathologists and the Leeds Pathology Protocol that all patients requiring pancreaticoduodenectomy for carcinoma should undergo this examination. This standardized specimen analysis allows consistent comparisons between series and, consequently, between pathologies. A recent study from Leeds [6] specifically assessed DC margins, but only included patients who underwent surgery before 2009. To date, there is no report of a series study that has conducted a resection margin comparison using this protocol, in a recent cohort.

Therefore, to compare the frequency and location of incomplete resections, the present study evaluated the resection margin status among patients who underwent pancreaticoduodenectomy for DC or PDAC, using standardized surgery and specimen protocol analysis. Furthermore, in the absence of international consensus and as it was previously reported that the 1mm clearance is an independent determinant of postresection survival [5], we focused the R1 status on that cut-off.

## 2. Material and Methods

### 2.1. Patient Selection

From 1 January 2010 to 31 December 2018, 355 consecutive patients who underwent pancreaticoduodenectomy with curative intent for PDAC (*n* = 288) or DC (*n* = 67) at the Institut Paoli-Calmettes, Marseille, France were evaluated prospectively, and followed-up until 15 April 2021. Patients with duodenal adenocarcinoma, ampulla adenocarcinoma, pancreatic cystic tumors, neuroendocrine tumors, benign tumors, or uncertain distinction between PDAC and DC were excluded. Data were entered into a clinical database. The study protocol was approved by the institutional review board of the Institute. Participants were informed about the study and provided written consent. Their data are contained in the investigators’ declared prospective institutional database (NCT02871336; CNIL n°Sy50955016U).

### 2.2. Surgery and Specimen Management

Pancreaticoduodenectomy was performed via a subcostal incision. First, a thorough abdominal exploration was performed. Contraindications for resection included intraoperative, histologically proven carcinomatosis, liver metastasis, and para-aortic lymph node metastasis in patients with PDAC [7]. Invasion of the superior mesenteric artery, celiac axis, or hepatic artery was not considered a contraindication for resection in selected patients (those who were physically fit, had an objective response to induction treatment, with long (>6 months) follow-up time before surgery with no metastasis detected, low (<500 IU/mL) carbohydrate antigen [CA] 19-9 serum level, and no neoadjuvant chemoradiation). Venous resection, if required, was performed as previously described [8], and the venous segment was clearly identified on the specimen by the surgeon. Pancreatic neck transection margins and common bile/hepatic duct transection margins were evaluated using frozen section analysis. If results were positive, either additional tissue was resected, or a total pancreatectomy was performed. All specimens were routinely inked by the surgeon in the operating room to facilitate margin assessment. The venous, arterial, and retroperitoneal margins were inked in blue, red, and yellow, respectively (Figure 1).

## 3. PDAC, Pancreatic Ductal Adenocarcinoma

### 3.1. Pathologic Analysis and Margin Assessment

All specimens were evaluated by a team of experienced pathologists to ensure standardized examination of the surgical specimens and relevant margin assessments. The precise origin of a tumor in the pancreatic head is, at times, difficult to determine (Figure 2). The macroscopic examination provides some evidence for determining the tumor origin; therefore, the location of the tumor epicenter should be assessed. Common bile duct tumors arise along the common bile duct route in the posterior-cranial aspect of the pancreatic head, above or at the level of the ampulla, and often involve posterior pancreatic margins. Pancreatic tumors can occur in any part of the pancreatic head, and their precise origin may be difficult to determine, particularly when they are large and involve more than one potential original site. In this study, microscopic confirmation of tumor origin was helpful in some cases, particularly when a precursor lesion (pancreatic or biliary intraepithelial neoplasia) was identified. It is important to remember that there are no immunohistochemical markers that distinguish between PDAC and bile duct carcinoma. Thus, precise determination of whether the tumors arose in the pancreatic or biliary ducts in this investigation was based on macroscopic and microscopic assessment.

Serial slicing of pancreatic head specimens was performed according to guidelines of the Royal College of Pathologists and the Leeds Pathology Protocol [2]. This allowed a precise millimetric study of each inked margin. Relationships between the tumor and the following specimen surfaces and margins were assessed: transection margins of the pancreatic neck; common bile duct; stomach; posterior surface (posterior margin, yellow ink); the margin toward the superior mesenteric artery (arterial margin, red ink); and the surface of the superior mesenteric vein/portal vein groove (venous margin, blue ink). Margin involvement (R1) was affirmed if tumor cells were present at the resection margin (0 mm), within 1 mm [5,9], or within 1.5 mm [5,10] from at least one inked margin. They did not include bile duct and pancreatic neck transection margins. The impact of the latter two definitive margin resection statuses on survival seems to be low [11,12], as opposed to what was observed in a study utilizing intraoperative frozen section analysis [13]. Patients who underwent venous resection were considered to have positive venous margins if the tumor was present at the resection margin (not intraluminally) (Figure 1). The pathological protocol also included maximal transverse diameter of the tumor, tumor-node-metastasis (TNM) according to the American Joint Committee on Cancer (AJCC, 7th edition), carcinoma differentiation grade, presence, or absence of perineural and lymphovascular invasions, and the number of lymph nodes retrieved from the specimen.

### 3.2. Study Parameters

Numerous clinical variables were evaluated. These included age, sex, body mass index, serum CA 19-9 level (U/mL; at diagnosis after jaundice resolution), biliary stenting, and administration of neoadjuvant treatment. Data on type of surgery (pancreaticoduodenectomy or total pancreatectomy), vascular resection, and enlarged resection (not including gallbladder and spleen) were also noted. Information regarding morbidities according to the Clavien–Dindo classification was recorded [14], including postoperative pancreatic fistula [15], hemorrhage, postoperative (in-hospital or 30-day) and 90-day mortality rates, perioperative red blood cell (RBC) transfusion rate (from surgery to home discharge), length of hospital stay, and adjuvant treatment administration. No patients were lost to follow-up. Censored data corresponded to patients alive at the date of censoring.

### 3.3. Statistical Analysis

Categorical variables were compared using the Fisher’s exact test or the chi-squared test, and continuous variables were compared using the student’s *t*-test. Overall survival (OS) was assessed using the Kaplan–Meier method (based on the date of diagnosis and the censoring date, 15 April 2021). Survival curves were compared using the log-rank test, and results were reported as hazard ratios (HRs) with 95% confidence intervals (CIs). Multivariate analysis was performed using stepwise logistic regression, integrating factors identified in the univariate analysis with a significance level of *p* < 0.1, unless they were highly clinically significant. Data analyses were performed using GraphPad Prism version 8 (GraphPad Software Inc., La Jolla, CA, USA) and SPSS^®^ version 24 (IBM Corp., Armonk, N.Y., USA). Statistical significance was set at *p* < 0.05.

## 4. Results

### 4.1. Patient Characteristics, Surgery, and Postoperative Courses

Patients with DC were mostly men (64% vs. 49%, *p* = 0.028), of older age (68 vs. 65 years, *p* = 0.033), required biliary stenting more frequently (93% vs. 77%, *p* < 0.01), and received a lower amount of neoadjuvant treatment (*p* < 0.001) than those with PDAC. The venous resection rate was higher among patients with PDAC (112/288, 39% vs. 10/67, 16%; *p* = 0.028). For the entire series, postoperative and 90-day mortality rates were both 1.5% for patients with DC and 1.7% and 2.4%, respectively, for those with PDAC. Patients with DC had higher incidence of pancreatic fistula (27/67, 40% vs. 52/288, 18%; *p* < 0.001) and perioperative RBC transfusion (24/67, 36% vs. 68/288, 24%; *p* = 0.04) than patients with PDAC. Postoperative and 90-day mortality rates were comparable. No difference was observed in adjuvant treatment administration between the two groups (Table 1).

### 4.2. Pathological Data 

Misdiagnosis preceding pathology was the case in 13 out of 288 patients with PDAC (4.5%), and in 3 out of 67 patients with DC (4.4%). Patients with PDAC had larger tumors (28.6 mm vs. 24 mm, *p* = 0.01) than patients with DC. The following were comparable between the two groups: number of lymph nodes examined, lymph node invasion rate, perineural and lymphovascular invasion rates, final invasion rates of the pancreatic neck and bile transection margins, and T stages. The R1 resection rate was comparable between patients with PDAC and those with DC, regardless of clearance margin (0 mm, <1 mm, or <1.5 mm). Among the three types of resection margins, the venous type was more frequently observed in both groups. Patients with PDAC had lower venous (3.6 mm vs. 7.7 mm, *p* = 0.001) and arterial (5 mm vs. 8.7 mm, *p* = 0.001) margin clearances than patients with DC. Retroperitoneal margin clearance and numbers of involved margins (1, 2, or 3) were comparable between DC and PDAC (Table 2).

### 4.3. Impact of Resection Margin on Survival 

Patients with DC had higher median (64 vs. 32 months) and 5-year (57% vs. 29%) survival rates (HR = 0.65, 95% CI 0.48–0.88, *p* = 0.012) than patients with PDAC (Figure 3). The R1 resection status using a cut-off value of 1 mm (those with clearances of 0 and 1.5 mm were also evaluated and not significant) did not impact the OS among patients with DC (median survival: 43 vs. 63 months; HR = 1.64, 95% CI 0.8–3.4, *p* = 0.13) (Figure 4a). However, in those patients with PDAC, R1-1mm resection status resulted in lower OS (median survival 23 months vs. 36 months; HR = 1.43, 95% CI 1.07–1,9 *p* = 0.01) (Figure 4b). In multivariate analysis, R1-1mm resection margin did not impact OS in patients with either PDAC or DC (Table 3).

## 5. Discussion

In this large monocentric study reflecting the current landscape of a high-volume surgery center, we used standardized specimen protocol analysis to show that patients with DC and PDAC had similar R1 resection rates, which seemed to impact OS more in PDAC than in DC.

### 5.1. Patient Characteristics, Surgery, and Postoperative Course

DC and PDAC are two different pathological entities with different clinical presentations. On one hand, because DC occurs in the bile duct lumen, jaundice was the most frequent symptom that led to a diagnosis. This also explains the higher biliary stenting rate observed among patients with DC. On the other hand, delayed diagnosis in patients with PDAC frequently leads to more advanced disease, which explains the higher level of neoadjuvant treatment and venous resection rates in our study patients, consistent with what has been shown in the literature [9,16].

Similarly, patients with DC are more likely to have soft pancreatic parenchyma, which explains the higher incidence of pancreatic fistula and associated RBC transfusion rates in these patients than what we observed in patients with PDAC. Nevertheless, the morbidity and mortality rates we present in this study are comparable and consistent with those reported at other high-volume surgery centers [16,17].

### 5.2. Pathologic Data

An earlier disease diagnosis probably explains why patients with DC had smaller tumor sizes than patients with PDAC. However, this did not appear to impact other pathologic data, as we did not observe differences in T stage, tumor grading, lymph node invasion, and perivascular or perineural invasion between the diseases. Similarly, we did not observe any significant differences according to the R1 resection rate, regardless of the clearance adopted and whether or not venous and arterial margin clearances were smaller among patients with PDAC. As previously reported [5], the venous groove was the most frequently invaded margin, comparable between the two diseases.

The recent study by Ethun et al. involving 1464 patients [1] showed that patients with PDAC were exposed to higher R1 resection rates than those with DC. Our finding was contradictory, and the validity of this could be questioned on the basis that our study had a smaller sample size (355 patients). However, our results should be taken into consideration because of the following three points. First, our study was specifically designed for and focused on margin resection assessment. This gives it a stronger axis when compared with that provided by the Ethus et al. study, which was dedicated to the overall comparison of the two diseases. Second, the monocentric characteristics of our series might be seen as a limitation, but it could also be considered as a strength, given the consistency found within our high-volume center. Indeed, our smaller but more surgically and pathologically homogenous sample should be considered in light of the patients coming from 13 institutions in the larger Ethus et al. study. Third, the period of inclusion was dramatically different between the two investigations (2000–2015 in Ethun et al. vs. 2010–2018 in our series). The longer enrolment period in the earlier study included a timeframe of crucial developments that occurred in pancreaticoduodenectomy specimen analyses in the late 2000s [2]. We posit that the specimen analysis in our investigation was more consistent and relevant because it was performed according to an unchanged standardized protocol. Furthermore, the comparable R1 resection rate we observed, particularly in the venous groove, makes sense, since the extra-pancreatic DC part has been recently highlighted [6] as favoring venous involvement and worsening prognosis.

We also wish to bring up the confusing role that neoadjuvant treatment might have played in the margin assessment, because it is more frequently used in patients with PDAC. Indeed, as neoadjuvant therapy kills tumor cells, residual cancer often consists of scattered tumor foci separated by stretches of non-neoplastic tissue [18]. Thus, the distance between remaining tumor cells increases, and the evaluation of residual tumor beyond the resection margin becomes more problematic. Despite this limitation, we were able to show that pancreaticoduodenectomy performed on patients with PDAC or DC resulted in comparable R1 resection rates.

### 5.3. Survival

Our data reinforces the reported favorable prognosis seen in patients with DC as opposed to those with PDAC [1], likely due to differences in tumor biology [19] and behavior [20]. Furthermore, we found that R1 resection affected survival among patients with PDAC but did not reach statistical significance in those with DC. We could argue that survival in patients with DC was more linked to biological aggressiveness of the tumor, as estimated by lymph node ratio [21], bilirubin adjusted serum CA 19-9 level [22], or platelet to lymphocyte ratio [23], rather than to resection margin. In contrast, the lack of impact of resection margin on survival may have been due to the smaller sample of patients with DC in our study. This could have had an influence on statistical power that might have resulted in a type II error. However, a recent meta-analysis reported in the literature [24] suggested that an increase in the number of study patients resulted in decreased precision and accuracy of pathological examinations, which was crucial in evaluating margins.

### 5.4. Strengths and Limitations

Study limitations include not using the 8th edition of the TNM AJCC-UICC staging system, which was not established in practice at the initiation of our study. In addition, there was an absence of a precise distinction between extra and intra-pancreatic DC parts. The unicentric character could also be seen as a strength that permitted both standardized resection and a pathological protocol. Additional strengths of our study include the prospective evaluation, completed during a recent time period. Importantly, we included a homogeneous comparative population with PDAC, as well as long-term survival assessment with sufficient follow up (at least two years and a half after surgery).

## 6. Conclusions

In conclusion, standardized specimen protocol analysis indicated that the R1 resection rate was comparable among patients who underwent pancreaticoduodenectomy for PDAC or DC. However, the R1-1mm resection margin was found to adversely impact OS in patients with PDAC, while patients with DC were confirmed to have a more favorable prognosis. An international consensus for prognosis-based R1 is needed to achieve comparability for biliary and pancreatic cancer.

## Figures and Tables

**Figure 1 jcm-10-03247-f001:**
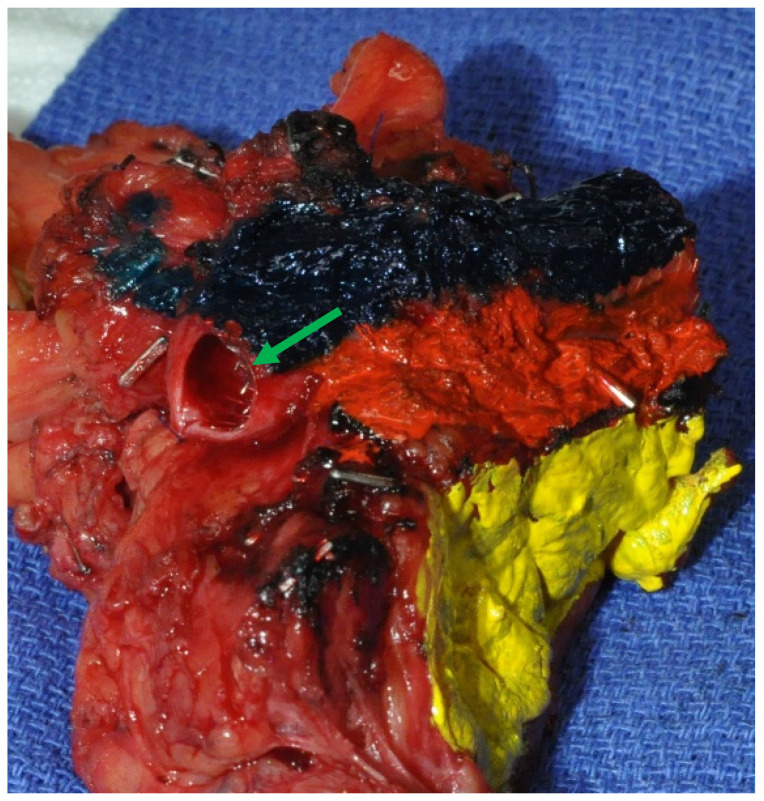
Pancreaticoduodenectomy specimen with long (8 cm) portal vein resection in a patient with locally advanced PDAC who received Folfirinox induction. A metallic stent (green arrow) was inserted preoperatively into the portal vein because of complete stenosis. The specimen was inked by the surgeon in the operating room; the venous, arterial, and retroperitoneal margins were inked in blue, red, and yellow, respectively.

**Figure 2 jcm-10-03247-f002:**
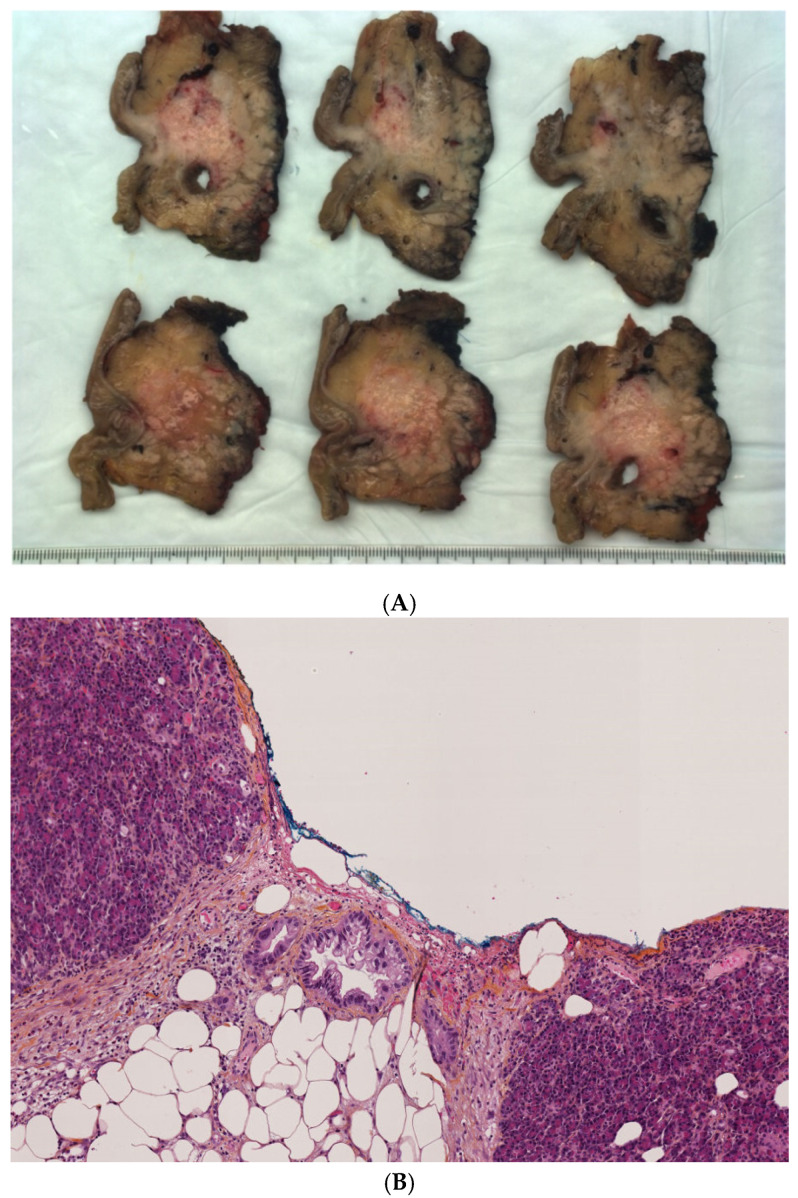
Pathologic examination for differential diagnosis between pancreatic adenocarcinoma and distal cholangiocarcinoma after pancreaticoduodenectomy. (**A**) Serial slicing, macroscopic view. Black arrow indicate the tumor; dot line arrow indicate the biliary tract; circle indicate the ampullary. (**B**) Hematoxylin and eosin staining of the same patient sample, microscopic view. Black arrow indicate the positive margin (in blue ink). The distance with adenocarcinoma gland is within 1 mm. Dot line arrow indicate the adenocarcinoma glandular cell.

**Figure 3 jcm-10-03247-f003:**
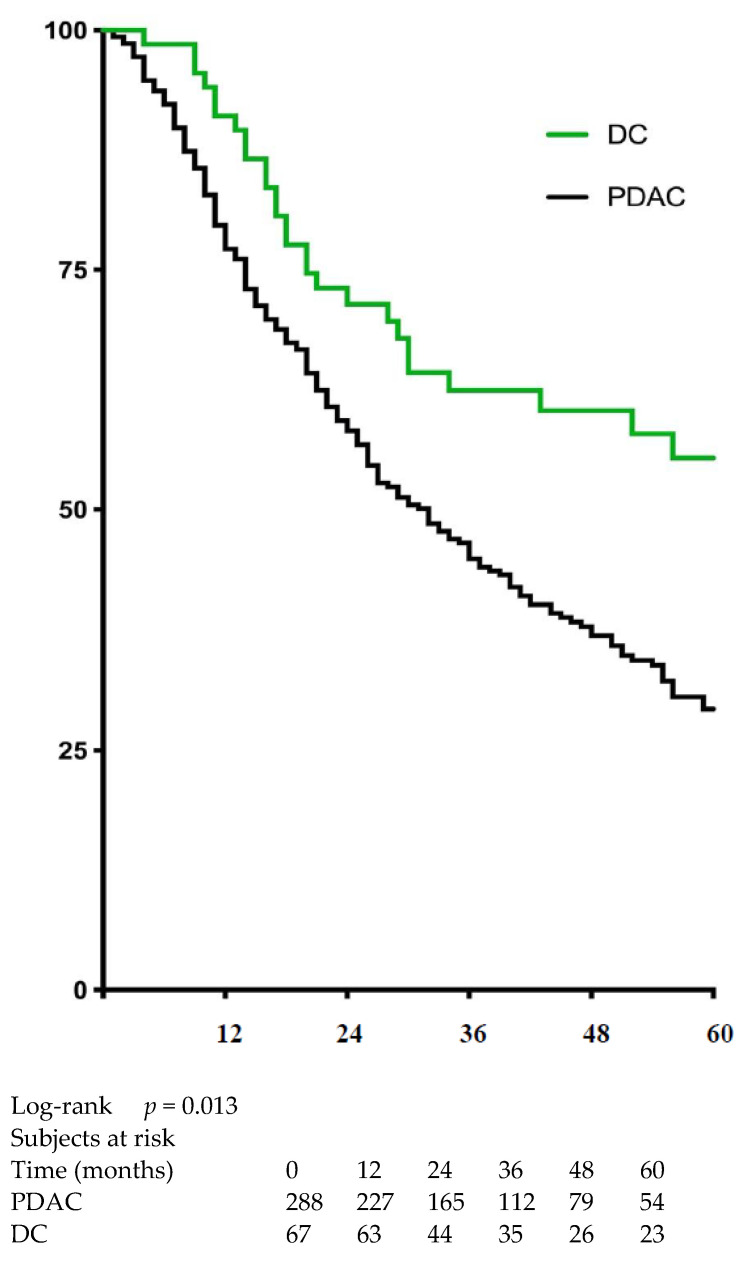
Overall survival among patients who underwent resection for distal cholangiocarcinoma (DC) or pancreatic ductal adenocarcinoma (PDAC).

**Figure 4 jcm-10-03247-f004:**
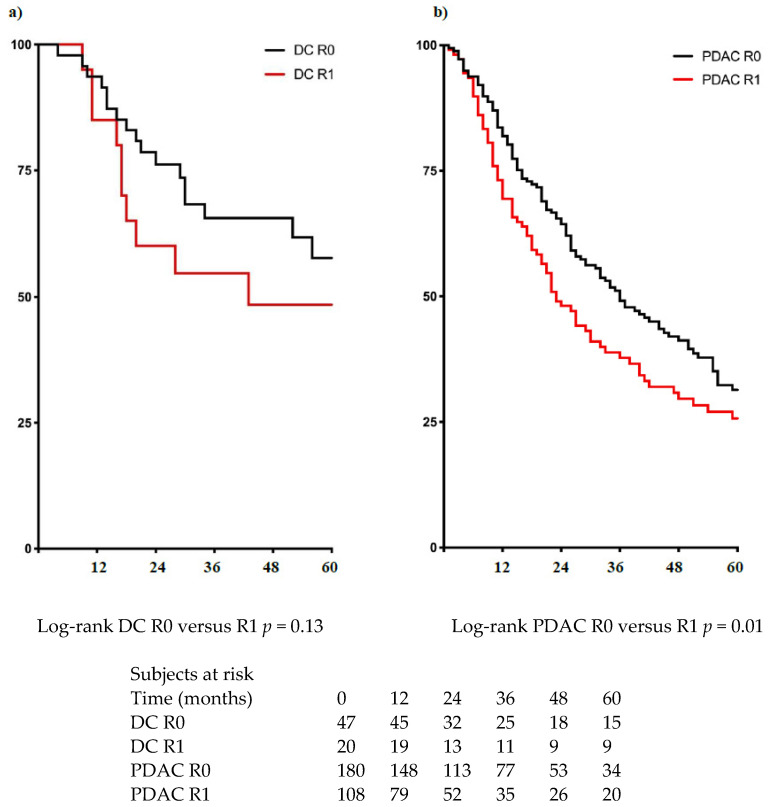
Overall survival among patients who underwent tumor resection according to margin status. (**a**) Distal cholangiocarcinoma (DC); (**b**) Pancreatic ductal adenocarcinoma (PDAC).

**Table 1 jcm-10-03247-t001:** Comparisons of baseline demographics, operative data, and postoperative courses between patients with distal cholangiocarcinoma (DC) and those with pancreatic ductal adenocarcinoma (PDAC).

	PDAC	DC	*p*-Value
*N*	288	67	-
Sex ratio (M/F)	0.97	1.8	0.028
Median age (range)	65 (29–86)	68 (52–81)	0.033
Mean BMI (±SD)	24.1 (±4.03)	24.3(±4.23)	0.81
Mean serum CA 19-9 level ^a^ (UI) (±SD)	438 (±1056)	164 (±365)	0.13
Biliary stenting (%)	220 (77)	62 (93)	<0.01
Neoadjuvant treatment (%)	123 (43)	3 (4.5)	<0.001
Total pancreatectomy (%)	25 (8.7)	2 (3)	0.11
Vascular resection (%)			
Venous resection	112 (39)	10 (15)	<0.001
Arterial resection	16 (5.6)	2 (3)	0.54
Enlarged resection (%)	8 (2.8)	2 (3)	1
Perioperative RBC transfusion (%)	68 (24)	24 (36)	0.04
Reintervention (%)	19 (6.6)	3 (4.5)	0.78
Morbidity rate (%)			
Overall	142 (49)	34 (51)	0.83
Clavien–Dindo grade 3–5	32 (11)	13 (19)	0.07
Mortality rate (%)			
30-day	5 (1.7)	1 (1.5)	1
90-day	7 (2.4)	1 (1.5)	1
Postoperative pancreatic fistula ^b^ (%)	52 (20)	27 (42)	<0.001
Biliary fistula (%)	5 (1.7)	1 (1.5)	1
Hemorrhage (%)	17 (5.9)	5 (7.5)	0.58
Median length of hospital stays (days) (range)	18 (8–83)	19 (10–41)	0.12
Adjuvant treatment (%)	195 (68)	38 (57)	0.63

BMI, body mass index; SD, standard deviation; CA, cancer antigen. ^a^ at diagnosis and after jaundice resolution. ^b^ calculated among patients who underwent pancreaticoduodenectomy.

**Table 2 jcm-10-03247-t002:** Comparisons of pathological features between distal cholangiocarcinoma (DC) and pancreatic ductal adenocarcinoma (PDAC).

	PDAC	DC	*p*-Value
Mean tumor size (mm) (±SD)	28.6 (±13.2)	24 (±12.4)	<0.01
T 3/4 stage (%)	185 (64)	39 (58)	0.36
Median number of examined lymph nodes (range)	16 (2–42)	15 (5–33)	0.4
Lymph node invasion (N+) (%)	186 (65)	40 (60)	0.45
Perineural invasion (%)	210 (73)	54 (81)	0.19
Lymphovascular invasion (%)	151 (52)	34 (51)	0.89
Carcinoma differentiation grade (%)			
Low	35 (12)	10 (15)	0.66
Intermediate	134 (47)	33 (49)	
High	119 (41)	24 (36)	
Final positive pancreatic neck margin ^a^ (%)	11 (3.8)	4 (6.2)	0.5
Final positive biliary transection margin (%)	8 (4.2)	4 (6)	0.25
R1 resection on inked margin (%)			
0 mm clearance	33 (11.5)	5 (7.5)	0.5
<1 mm clearance ^b^	109 (38)	20 (30)	0.22
<1.5 mm clearance	142 (49.3)	25 (37.3)	0.08
Location of R1 margin (%)			
Venous margin	67 (23)	11 (16)	0.22
Arterial margin	36 (12)	7 (10)	0.64
Retroperitoneal margin	52 (18)	10 (15)	0.54
Mean (mm) (±SD)/median (mm) margin clearance			
Venous margin	3.57 (±3.99)/6	7.74 (±6.6)/2	<0.001
Arterial margin	5.03 (±4.7)/7	8.74 (±6.86)/4	<0.001
Retroperitoneal margin	5.37 (±5.6)/5	6.79 (±6.14)/4	0.09
Number of involved inked-margins (%)			
1	63 (22)	13 (19)	0.66
2	39 (14)	6 (9)	0.31
3	7 (2.4)	1 (1.5)	1

^a^ calculated among patients who underwent pancreaticoduodenectomy. ^b^ Retained clearance for the present study.

**Table 3 jcm-10-03247-t003:** Univariate and multivariate analyses of factors influencing the overall survival among patients with distal cholangiocarcinoma (DC) or pancreatic ductal adenocarcinoma (PDAC).

	Univariate *p*-Value	OR (95% CI)	Multivariate *p*-Value
Neoadjuvant treatment			
PDAC	0.47	1.68 (1.14–2.48)	<0.01
DC	0.24	-	-
Perioperative RBC transfusion			
PDAC	0.016	1.68 (1.12–2.51)	0.011
DC	0.48	-	-
Adjuvant treatment			
PDAC	0.01	0.533 (0.366–0.778)	<0.001
DC	0.57	-	-
T 3/4 stage			
PDAC	<0.001	4.47 (2.65–7.55)	<0.001
DC	<0.001	4.45 (1.44–13.8)	<0.01
Lymph node invasion (N+)			
PDAC	<0.001	2.35 (1.41–3.90)	<0.01
DC	0.013	2.06 (0.687–6.18)	0.2
Perineural invasion			
PDAC	<0.001	1.69 (1.00–2.85)	0.049
DC	0.026	3.38 (0.737–15.5)	0.12
Differentiation grade			
Low vs intermediate PDAC	<0.01	0.704 (0.401–1.23)	0.22
Low vs. high PDAC	0.6	0.551 (0.304–0.997)	0.049
DC	-	-	-
R1 resection inked-margin (1 mm clearance)			
PDAC	<0.01	0.889 (0.496–1.59)	0.69
DC	0.22	-	-
Venous location of R1 margin			
PDAC	<0.001	1.70 (0.974–2.98)	0.062
DC	0.38	-	-
Bile duct location of R1 margin			
PDAC	0.069	-	-
DC	0.024	2.91 (0.628–13.5)	0.17
Number of involved inked-margins			
2 in PDAC	<0.001	1.45 (0.866–2.43)	0.16
3 in PDAC	0.22	2.57 (0.967–6.80)	0.058
2 in DC	0.071	-	-
3 in DC	0.46	-	-

OR, odds ratio; CI, confidence interval.

## Data Availability

The datasets generated and/or analyzed during the current study are not publicly available due to patient privacy concerns but are available from the corresponding author on reasonable request.

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
