# Peer review of "Prospective Evaluation of Resection Margins Using Standardized Specimen Protocol Analysis among Patients with Distal Cholangiocarcinoma and Pancreatic Ductal Adenocarcinoma"

_jcm, 2021, doi:10.3390/jcm10153247_

Round 1

Reviewer 1 Report

This study aimed to evaluate the resection margin status of patients who underwent resection of distal cholangiocarcinoma and pancreatic ductal adenocarcinoma. Authors collected the data prospectively, particularly they collected the pathologic margin status carefully and vigorously. And, they evaluated not only the transection resection margin of the duct but also circumferential margins including venous, arterial and lymphovascular margin. This is commendable. And the impact of the margin status of both distal bile duct cancer and pancreatic head cancer, whether it is R0 or R1, on the survival was main focus of this study.   

Criticisms:

  1. Generally, R0 resection means that there is no residual tumor on the resection margin, and R1 means that there is cancer remains in the resection margin. When I looked up the table 2, authors evaluated the R1 margin status and classified it to 0mm clearance, <1mm clearance and <1.5mm clearance. R1 is to be regarded as only for 0mm margin of cancer. Why authors 1mm or 1.5 mm clearance margin included in the R1 resection group. When reviewer looked up the Royal College of Pathology and the Leeds Pathology Protocol(Ref 4), there is no guideline about the criteria of R1. Is not only ‘0 clearance margin’ R1? If this is correct, it should be revised in the table and text. If the definition of R1 is not consistent to other papers, the data across entire study is to be not consistent with other studies.  If it is correct, it should be reanalyzed. Authors should show the generally accepted definition or guideline of R1 status and authors’ opinion.
  2. Reviewer also can’t understand the exact meaning or difference of the final positive pancreatic and biliary transection margins, R1, location of R1 margin and number of involved margin in the table 2. Each data are different form each other.
  3. The 5 year survival rate of both DC and PDAC shows excellent result in the Fig. 3. Reviewer interested in the regimens and response rate of neo-adjuvant chemotherapy, and the type of regimens used for adjuvant as well as neo-adjuvant chemotherapy.

Author Response

  1. Generally, R0 resection means that there is no residual tumor on the resection margin, and R1 means that there is cancer remains in the resection margin. When I looked up the table 2, authors evaluated the R1 margin status and classified it to 0mm clearance, <1mm clearance and <1.5mm clearance. R1 is to be regarded as only for 0mm margin of cancer. Why authors 1mm or 1.5 mm clearance margin included in the R1 resection group. When reviewer looked up the Royal College of Pathology and the Leeds Pathology Protocol(Ref 4), there is no guideline about the criteria of R1. Is not only ‘0 clearance margin’ R1? If this is correct, it should be revised in the table and text. If the definition of R1 is not consistent to other papers, the data across entire study is to be not consistent with other studies. If it is correct, it should be reanalyzed. Authors should show the generally accepted definition or guideline of R1 status and authors’ opinion.

Response:

Thank you very much for your valuable comments, thoughtful suggestions, and insights. We previously highlighted that a more restrictive definition for resection margins, using a predefined cut-off value of 1 mm, was chosen; however, clearances of 0 and 1.5 mm were also evaluated.  Furthermore, two studies by the UK Royal College of Pathologist defined microscopic residual tumor as the presence of tumor cells within 1.0 mm of the margins(The Royal College of Pathologists. London. Dataset for the histopathological reporting of carcinomas of the pancreas, ampulla of Vater and common bile duct. Available at: http://www.rcpath.org. 2017. Survey of UK histopathologists’ approach to the reporting of resection specimens for carcinomas of the pancreatic head. Feakin et al, J Clin Pathol 2013).  This is supported by recent papers from our group and multicenter studies showing that a 1-mm R1 definition is preferable to better evaluate the impact on survival, as it is an independent determinant of post-resection survival (Prognostic value of resection margin involvement after pancreaticoduodenectomy for ductal adenocarcinoma: updates from a French prospective multicentre study. Delpero et al, Ann Surg 2017. https://doi.org/10.1097/SLA.0000000000002432. Pancreaticoduodenectomy for pancreatic ductal adenocarcinoma: a French multicentre prospective evaluation of resection margins in 150 evaluable specimens. Delpero et al, HPB 2013 https://doi.org/10.1111/hpb.12061). We have updated the Introduction section accordingly to justify the R1 choice. The respective changes are highlighted in yellow.

  1. Reviewer also can’t understand the exact meaning or difference of the final positive pancreatic and biliary transection margins, R1, location of R1 margin and number of involved margin in the table Each data are different form each other.

Response:

Thank you for pointing this out. We apologize for the lack of clarity. We evaluated the relationships between the tumor and the following specimen surfaces and margins, and assessed the transection margins of the pancreatic neck, common bile duct, and posterior surface of the stomach (posterior margin, yellow ink); the margin toward the superior mesenteric artery (arterial margin, red ink); and the surface of the superior mesenteric vein/portal vein groove (venous margin, blue ink). We did not include the stomach, bile duct, and pancreatic neck transection margins in the R1 analysis, as the impact of these definitive margin resections on survival appeared low (Value of intraoperative neck margin analysis during whipple for pancreatic adenocarcinoma: A multicenter analysis of 1399 patients. Kooby et al, Ann Surg 2014. Prognostic comparison of the longitudinal margin status in distal bile duct cancer: R0 on first bile duct resection versus R0 after additional resection. J Hepato-Biliary-Pancreat Sci Park et al 2019) when compared to observations from a study utilizing intraoperative frozen section analysis (Revision of pancreatic neck margins based on intraoperative frozen section analysis is associated with improved survival in patients undergoing pancreatectomy for ductal adenocarcinoma. Zhang et al, Ann Surg 2019). Thus, margin involvement (R1) was confirmed by the presence of tumor cells in the margin of resection at a minimum of one of the inked margins. We have updated Tables 2 and 3 accordingly to clarify this purpose.

  1. The 5 year survival rate of both DC and PDAC shows excellent result in the Fig. 3. Reviewer interested in the regimens and response rate of neo-adjuvant chemotherapy, and the type of regimens used for adjuvant as well as neo-adjuvant chemotherapy.

Response:

Considering the period of inclusion for this study (January 1, 2010, to December 31, 2018), the chemotherapy regimens were as follows:

  • For PDAC, Folfirinox for neoadjuvant treatment (43%), along with gemcitabine or with gemcitabine and capecitabine during the last year, in the adjuvant setting (68%). The relatively good 5-year survival is a selection bias, as all patients were fit for surgery and resected.
  • For DC, gemcitabine and cisplatin were administered in the rarely neoadjuvant treatment setting (4.5%) and capecitabine was used for adjuvant treatment (57%).

Despite the low efficiency of chemotherapy for DC, the 5-year survival was interesting (57%), as you highlighted, and was one of the interests of this study, as comparative and focused studies on DC are scarce.

Concerning the response rate of the neo-adjuvant treatment, unfortunately, our study was not designed to answer this question, but we report this information in another study from our institution. The resectability rate was approximately 38% after Folfirinox neoadjuvant treatment (Borderline or locally advanced pancreatic adenocarcinoma: A single center experience on the FOLFIRINOX induction regimen. Garnier et al, EJSO 2020).

Reviewer 2 Report

GENERAL COMMENTS:

The paper presents an elegant piece of research concerning the role of resection margin status in patients with distal cholangiocarcinoma and pancreatic ductal adenocarcinoma. The Royal College of Pathologists and the Leeds Pathology Protocol guidelines were used in the patient assessment. The authors have done a tremendous amount of work. Nevertheless, some issues have to be addressed.

SPECIFIC COMMENTS:

Line 73- “select patients” change to selected patients;

Line 106- “difficult diagnosis” should be changed to differential diagnosis;

Line 111- “the ampullary” change to the ampulla;

Line 116- “the adenocarcinoma glandular cell” should be the adenocarcinoma glandular cells;

Line 133- differentiation grade change to carcinoma differentiation grade;

Line 143- change to red blood cell transfusion rate;

Line 163- change to …mortality rates were both 1.5% for patients…

Line 166- change to perioperative RBC transfusion;

Table 1- change to perioperative RBC transfusion;

Table 2- “grade” change to carcinoma differentiation grade;

Fig. 4 b- what does the ADK abbreviation mean?

Table 3- why PDAC neoadjuvant treatment with the univariate p=0.47 and 3 margins in PDAC with the univariate p=0.22 were included in multivariate logistic regression analysis?

I cannot understand the OR values for the overall survival presented for the following factors: adjuvant treatment (PDAC OR <1); T3/4 stage (for both disorders OR>1 and very high), lymph node, and perineural invasion (both PDAC OR >1 and high).

Line 253- change to RBC transfusion rate

Revision of the paper is required.

Author Response

Line 73- “select patients” change to selected patients;

Line 106- “difficult diagnosis” should be changed to differential diagnosis;

Line 111- “the ampullary” change to the ampulla;

Line 116- “the adenocarcinoma glandular cell” should be the adenocarcinoma glandular cells;

Line 133- differentiation grade change to carcinoma differentiation grade;

Line 143- change to red blood cell transfusion rate;

Line 163- change to …mortality rates were both 1.5% for patients…

Line 166- change to perioperative RBC transfusion;

Table 1- change to perioperative RBC transfusion;

Table 2- “grade” change to carcinoma differentiation grade;

Line 253- change to RBC transfusion rate

Response:

Thank you very much for your valuable comments, thoughtful suggestions, and insights. We have updated the  different sections accordingly, and the changes are highlighted in yellow.

Fig. 4 b- what does the ADK abbreviation mean?

Response:

Thank you for pointing this out. We apologize for this error. We replaced ADK with PDAC, as shown in Fig. 4b.

Table 3- why PDAC neoadjuvant treatment with the univariate p=0.47 and 3 margins in PDAC with the univariate p=0.22 were included in multivariate logistic regression analysis?

I cannot understand the OR values for the overall survival presented for the following factors: adjuvant treatment (PDAC OR <1); T3/4 stage (for both disorders OR>1 and very high), lymph node, and perineural invasion (both PDAC OR >1 and high).

Response:

As specified in the Methods section, the multivariate analysis was performed using stepwise logistic regression, integrating factors identified in the univariate analysis with a significance level of P<0.1, unless they were highly clinically significant. Thus, we included the clinical variable of interest as the neoadjuvant treatment and when all inked margins were involved (3), even if they were not significant in the univariate analysis. Furthermore, it did not change the global significance. Concerning the OR values, for the Cox regression analysis, an OR>1 was associated with a worse survival (death was the end point), and OR<1 was considered a “protective” factor.

Adjuvant treatment delivery was associated with better OS (OR<1), contrary to T3/T4 stage, lymph node involvement, and perineural invasion. The question remains for neoadjuvant treatment (OR>1), but there was a selection bias, as these patients had more advanced disease, and the study was not designed for comparison with neoadjuvant treatment.

Round 2

Reviewer 1 Report

Reviewers have shown the papers supporting authors’ method regarding criteria of R1 resection margin.

Reviewer recognized that there has been debated of the prognostic correlation with the resection margin clearance.

Reference 1: As margin clearance increased, R1 status became a more powerful independent predictor of outcome; however, margin clearance did not relate to site of tumor recurrence. These data demonstrate that margin clearance by at least 1.5 mm identifies a subgroup of patients which may potentially achieve long-term survival. This study further confirms the need to achieve standardization across pancreatic specimen reporting. Stratification of patients into future clinical trials based upon the degree of margin clearance may identify those patients likely to benefit from adjuvant therapy. (J Clin Oncol 27:2855-2862)

Reference 2: In the context of adjuvant therapy, the resection margin status remains an important independent determinant of postresection survival. R0/ R1 resection rates and associated survival vary significantly with the definitions used. An international consensus is urgently needed to achieve comparability with respect to studies and protocols on patients with adjuvant therapy (Ann Surg 2017;265:565–573)

Reference 3: Tumor clearance <1.0 or <1.5 mm was an independent determinants of postresection survival in certain subgroups. To avoid misinterpretation, future trials should specify the clearance margin in millimeter. (Ann Surg 2017 Nov;266(5):787-796)

In several studies published so far, they proposed urgent or future trials is necessary to define the R1 criteria in terms of prognostic relevance with R1 status, supposedly tumor free margin <1.5mm is to be defined as R1. However it was not defined internationally, For rectal cancer, this concept is adopted already, but not for the biliary-pancreatic cancer yet.

Author Response

We thank you for your thoughtful suggestions and insights, which have enriched the manuscript and produced a better and more balanced account of the research. We mitigated the conclusion and emphasized on the urgent need for international definition of prognostic based R1, as you highlighted the fact that is not adopted yet for biliary and pancreatic cancer. This modest work is just another brick in the wall in this way.